# Study on Detection and Recognition of Traffic Lights Based on Improved YOLOv4

**DOI:** 10.3390/s22207787

**Published:** 2022-10-13

**Authors:** Ying Zhao, Yiyuan Feng, Yueqiang Wang, Zhihan Zhang, Zhihao Zhang

**Affiliations:** 1College of Engineering and Technology, Southwest University, Chongqing 400715, China; 2Department of Autonomous Driving, Changan Research Institute of Automotive Engineering, Chongqing 400023, China

**Keywords:** YOLOv4, ShuffleNetv2, lightweight, improved k-means, CS^2^A attention mechanism, data augmentation, traffic lights recognition

## Abstract

To resolve the issues of a deep backbone network, a large model, slow reasoning speed on a mobile terminal, low detection accuracy for small targets and difficulties detecting and recognizing traffic lights in real time and accurately with YOLOv4, a traffic lights recognition method based on improved YOLOv4 is proposed. The lightweight ShuffleNetv2 network is utilized to replace the CSPDarkNet53 network of YOLOv4 to satisfy the requirements of a mobile terminal. The reformed k-means clustering algorithm is applied to generate anchor boxes for avoiding the sensitivity issue of outliers and initial values. A novel attention mechanism named CS^2^A is added to enhance the extraction capability of effective features. Multiple data augmentation methods are combined to improve the generalization ability of the model. Ultimately, the detection and recognition of traffic lights can be realized. The S^2^TLD dataset is selected for training and testing, and it can be proved that the recognition accuracy and model size are greatly optimized. Meanwhile, a self-made dataset is selected for training and testing. Compared with the conventional YOLOv4, the recognition accuracy of the proposed algorithm for traffic lights’ state information increases by 1.79%, and the model size decreases by 81.97%. Appropriate scenes are selected for real-vehicle testing and the results demonstrate that the detection speed of the presented algorithm increases by 16%, and the recognition effect for small targets increases by 37% in comparison with conventional YOLOv4.

## 1. Introduction

Nowadays, urban traffic networks are becoming more and more complex. To ensure autonomous vehicles can be integrated into urban transport safely it is essential for them to be able to recognize the state of traffic lights rapidly and accurately. Consequently, the detection and recognition of traffic lights have attracted more attention from research scholars [1,2]. Currently, the recognition of traffic lights is mainly found in image processing and deep learning [3,4].

Since conventional image processing approaches are very sensitive to environment, the robustness for various environmental illuminations is rather poor, which cannot be conducive to applying it further in practice [5,6,7]. Deep learning is a machine learning technology that is widely applied in fault detection and object recognition for rapid development of hardware computing ability [8,9,10]. The methods based on deep learning mainly adopt convolutional neural network (CNN) to learn feature information autonomously, which possesses excellent robustness for various environments and satisfies the needs of recognition in different environments. Xiong, H. et al. [11] presented a candidate region generation method for traffic lights based on genetic optimization, and a location and classification method for traffic lights found on deep neural network, which has a high recall rate for traffic lights and can effectively distinguish between different categories of traffic lights. Qian, H.Y. et al. [12] put forward a lightweight shuffleNetv2 backbone network based on YOLOv2 for traffic light detection and recognition. The YOLOv2 backbone neural network was replaced by shuffleNetv2 network. A batch normalization layer and a nonlinear activation function layer were added after each convolution layer, which accelerated the convergence speed of the model training process and avoided the occurrence of overfitting. Li, C.Y. et al. [13] proposed a simplified network based on YOLOv3, in which FNC, FPN and ResNet were retained while the number of parameters and residual layers of each layer decreased, and densely connected network space pyramid pooling was added. It solved the problem of the low detection speed of YOLOv3 on embedded platforms effectively. Based on automatic classification of OCT retinal lesion images, Chen, S.S. et al. [14] proposed a convolutional neural network GM-OCTnet with multi-channel, multi-scale and relatively lightweight characteristics. Wang, L. et al. [15] propounded a small-scale traffic lights detection model based on YOLOv3 algorithm, and the proposed detection model employed leapfrog feature fusion and k-means clustering algorithm to enhance detection effect of small-scale targets. Wang, Q.Y. et al. [16] proposed a traffic light detection and recognition method based on YOLOv4, which utilized shallow feature enhancement mechanism and bounding box uncertainty prediction mechanism to improve detection and recognition abilities for small targets. Wang, L.G. et al. [17] presented an LED-LeNet convolutional network recognition algorithm, which improved LeNet-5 network and recognition effects of the algorithm on digitals formed by LED lights in natural scenes by preprocessing the image through data augmentation, using Swish activation function and introducing Dropout regularization. Xu, Y.J. et al. [18] proposed a lightweight target detection network based on the design principle of the YOLO series single lens target detection network. The GhostModule module of GhostNet was incorporated, the Efficient Channel Attention (ECA) module was added in convolutional block, and the Distance-IOU loss was introduced, which effectively accelerated the convergence speed of the network and achieved lightweight design. Yu, G. et al. [19] suggested removing part of the original image background with the adoption of lane detection, separating red or green by using adaptive edge detection, and finally adopting TLRNet network to provide satisfactory results for multi-scale traffic lights on embedded platforms. Li, Y. et al. [20] utilized generative adversarial network for data augmentation and inserted Coordinate Attention (CA) mechanism to YOLOv4 to boost the recognition effect of the model on small features. In terms of traffic light recognition strategy and lightweight/small target, related main reviewed works are collected and listed in Table 1.

Based on the abovementioned pioneering research, a traffic lights detection and recognition method based on the improved YOLOv4 is presented herein.

●To resolve the issues of deep backbone network, large model and slow reasoning speed on a mobile terminal, the CSPDarkNet53 network of YOLOv4 is replaced by the lightweight ShuffleNetv2 network.●The improved k-means clustering algorithm is deployed to obtain anchor boxes to avoid the influences of outlier points on clustering effects and compensate for the sensitivity issue of initial clustering center of the original algorithm.●A novel attention mechanism named CS^2^A is inserted to enhance effective features and augment the ability of feature extraction.●Multiple data augmentation methods are utilized to enrich samples and ameliorate the generalization and robustness of the model.

The main content of this work contains eight sections. Section 2 introduces the traditional YOLOv4 algorithm. Section 3 explains the proposed algorithm theoretically. Section 4 presents the results of evaluation criteria. Section 5 gives the results of ablation test and comparison test on datasets. Section 6 examines the model complexity of each algorithm and carries out specific analyses. Section 7 presents the results of a real-vehicle test. Section 8 concludes this work.

## 2. YOLOv4

YOLOv4 is a target detection algorithm. The network structure is composed of backbone network and detection network, which are applied for feature extraction and multiscale prediction, respectively [21]. The overall network structure of YOLOv4 is described in Figure 1.

### 2.1. YOLOv4 Backbone Network

CSPDarkNet53 network is utilized as the backbone network of YOLOv4. The original 52 convolutional layers in DarkNet53 are retained in CSPDarkNet53, and the cross-stage feature fusion strategy is added in DarkNet53 [16]. Compared with DarkNet53 network, the number of parameters and the computation time of CSPDarkNet53 decrease significantly, and the feature extraction ability of the network is boosted by adding CSP structure to five Resblock_body [22].

Unlike DarkNet53, the Mish activation function is utilized in CSPDarkNet53. Mish is a smooth and non-monotonic activation function [23]. The expression is defined in Equation (1) and the corresponding function graph is also presented in Figure 2.
(1)$$F\left(x\right)=x\cdot \tanh(\mathrm{In}(1+e^{x}))$$

Compared with LeakyReLU activation function utilized in DarkNet53, the Mish activation function is equipped with an outstanding ability of generalization and can stabilize network gradient flow better and ameliorate the quality of results.

### 2.2. YOLOv4 Detection Network

The detection network of YOLOv4 can be divided into a feature fusion module and a prediction network module named YOLO HEAD. The feature fusion module mainly consists of an SPP spatial pyramid pooling module, and the PANet feature fusion network is deployed to acquire three reinforcing feature layers by fusing multi-scale features. The prediction network module is applied to predict the results.

The SPP spatial pyramid pooling module is applied to fix image size to equip the network for inputting images of arbitrary size and to avoid geometric distortion caused by cropping or stretching input image, which affects original features [24].

The PANet network is a feature fusion network based on Feature Pyramid Network (FPN) [25]. Large target features are obtained by constructing the up-sampling feature pyramid of deep features. Meanwhile, a down-sampling feature pyramid of shallow features is added to get small target features and improve a fusion effect of deep features and shallow features. The network structure of PANet is indicated in Figure 3.

Three enhanced feature layers are put into the prediction network module. One-time convolution of 3 × 3 is applied to integrate features. One-time convolution of 1 × 1 is used to adjust the number of channels. The prediction results are obtained by classification and regression in accordance with the number of classes in the training set.

## 3. Improved YOLOv4

Based on YOLOv4, an improved lightweight target detection network is proposed to address the issues of deep backbone network, large model, slow reasoning speed on a mobile terminal, low detection accuracy for small targets and difficulties in detecting and recognizing traffic lights in real time and accurately. According to the structure of the algorithm, the lightweight ShuffleNetv2 network is utilized to replace the CSPDarkNet53 network of YOLOv4. A novel attention mechanism based on CA attention mechanism is added after three effective feature layers of the output of the backbone network. In addition, the ameliorated k-means clustering algorithm is adopted to obtain appropriate anchor boxes. Multiple data augmentation methods are combined to enhance the generalization ability of model. Ultimately, the network model can meet the requirements of a mobile terminal while having better detection and recognition effects for small targets like traffic lights. The overall structure of improved YOLOv4 is demonstrated in Figure 4.

### 3.1. Backbone Network Improvement

The lightweight ShuffleNetv2 network is regarded as a backbone network in the framework of YOLOv4. The ShuffleNetv2 network, which was proposed by improving the ShuffleNetv1 network [26] with a large amount of experimental data, is a lightweight neural network designed for mobile terminals.

Two types of block unit for the ShuffleNetv2 are elaborated in Figure 5.

As stride is equal to 1, the block unit in Figure 5a is utilized to split the channel of input feature matrix. The left branch can be regarded as residual edge while two-time convolution of 1 × 1 and one-time DW convolution of 3 × 3 are adopted on the right branch. After the convolution, the channels are spliced by Concat. The number of output channels is consistent with that of input channels. The residual edge is not convolved in this block unit, which is deployed to deepen the network layer.

As stride is equal to 2, the block unit in Figure 5b is employed and the channel of input feature matrix cannot be split. One-time DW convolution of 3 × 3 and one-time convolution of 1 × 1 are adopted on the left branch. Two-time convolution of 1 × 1 and one-time DW convolution of 3 × 3 are adopted on the right branch. Following after the convolution, the channels are spliced by Concat. The number of output channels doubles. Both the left and right branches are convolved in this block unit to compress the feature layer for down-sampling.

Generally, ShuffleNetv2 is consistent with ShuffleNetv1 in the overall framework. To satisfy the four principles of lightweight network design propounded by Ma N et al. [27], a large number of groups are removed from ShuffleNetv1 to reduce intensive use of convolution and accelerate network reasoning speed. Meanwhile, a convolution layer of 1×1 is added before GlobalPool to avoid the loss of accuracy.

The overall structure of ShuffleNetv2 is described in Table 2.

### 3.2. Attention Mechanism

Human eyes can quickly scan a whole image to find the regions of interest (ROI) and then allocate attention to these regions. Finally, the ROI in an image can be perceived with high resolution with detailed information, while the surrounding background around it can be perceived with low resolution to suppress useless information [28]. By introducing an attention mechanism, the network can adaptively focus on more significant features in images and enhance the extraction of effective features.

As for the two-dimensional image put into CNN, the two dimensions stand for scale space of image and channel, respectively. Traditionally, an attention mechanism can be divided into channel attention mechanism, spatial attention mechanism and a combination of the two attention mechanisms. The channel attention mechanism focuses on the enhancement or suppression of the importance of different feature channels, spatial attention mechanism focuses on the enhancement or suppression of the importance of different image features, and spatial and channel fusion attention mechanism focuses on both.

Convolutional Block Attention Module (CBAM), which combines channel attention and spatial attention in parallel, can effectively increase the effect of target detection [29]. CA embeds position information in channel attention. It is a more suitable attention mechanism for lightweight networks as the effect is better than CBAM in lightweight networks [30]. Herein, more attention should be paid to target an image in the actual application of target detection, that is, spatial attention. In complex traffic environments with vehicle taillights and ambient streetlights, the network should pay more attention to traffic lights and suppress the importance of other environmental interference substances to achieve a better effect for traffic light detection and recognition. Therefore, based on CA, a new spatial and channel fusion attention mechanism named CS^2^A is proposed, which further fuses the spatial attention and channel attention fused with location information. The CS^2^A is a spatial and channel fusion attention mechanism that pays more attention to scale space. The principle of CS^2^A is presented in Figure 6.

#### 3.2.1. Channel Attention

The channel attention fused with location information is utilized as the new channel attention mechanism.

Average-pooling is applied in images along width and height directions, as indicated in Equations (2) and (3). (H,1) or (1,W) is deployed as pooling kernel.
(2)$$O_{1_{c}}(h)=\frac{1}{W}\sum_{0\leq i\leq W}I_{c}(h,i)$$
(3)$$O_{1_{c}}(w)=\frac{1}{H}\sum_{0\leq j\leq W}I_{c}(j,w)$$

Of which, $O_{1_{c}}(h)$ represents the output of *c*-channel with *h*-height. $O_{1_{c}}(w)$ stands for the output of *c*-channel with *w*-width. $I_{c}$ is the input of *c*-channel.

The above outputs are concatenated and the nonlinear activation function is introduced to obtain intermediate feature mapping of spatial information coding, as expressed in Equation (4).
(4)$$f=AF(F_{1}([O_{1_{c}}(h),O_{1_{c}}(w)]))$$
where, *AF* denotes nonlinear activation function; *F_1_* represents convolution transformation function of 1 × 1; and *f* signifies intermediate feature mapping. 

The tensor *f* can be adjusted to get the weights of feature points along two directions, Equations (5) and (6) are presented as follows.
(5)$$f^{\prime }{}^{h}=sigmoid(F_{h}(f(h)))$$
(6)$$f^{\prime }{}^{w}=sigmoid(F_{w}(f(w)))$$

Of which, $f^{\prime }{}^{h}$ and $f^{\prime }{}^{w}$ stand for the weight of feature points along height direction and the weight of feature points along width direction, respectively. Fh and Fw denote the convolution transform functions of 1 × 1 along height and width directions, respectively. 

Finally, the output of the new channel attention $O_{2_{c}}(i,j)$ can be characterized as Equation (7).
(7)$$O_{2_{c}}(i,j)=I_{c}(i,j)\times f_{c}{{}^{\prime }}^{h}(i)\times f_{c}{{}^{\prime }}^{w}(j)$$

#### 3.2.2. Spatial Attention

One-time maximum pooling and one-time average pooling are performed to the input channel attention feature layer $O_{2_{c}}(i,j)$. One-time convolution of 7 × 7 is applied to splice.
(8)$$f^{\prime \prime }=F_{7}([AvgPool(O_{2_{c}}),MaxPool(O_{2_{c}})])$$

Among which, *F_7_* stands for convolution transformation function of 7 × 7.

The sigmoid activation function can be introduced to get the weights of feature points, as represented in Equation (9).
(9)$$f^{\prime \prime \prime }=sigmoid(f^{\prime \prime })$$
where, $f^{\prime \prime \prime }$ donates the weight of each feature point.

Eventually, the output of spatial attention, that is, the output of the whole attention mechanism, is expressed as the following Equation (10).
(10)$$O_{3_{c}}(i,j)=O_{2_{c}}(i,j)\times f^{\prime \prime \prime }(i,j)$$

### 3.3. Anchor Boxes Generation

K-means clustering algorithm is utilized in YOLOv4 to obtain anchor boxes for the PASCAL VOC dataset [31]. As the PASCAL VOC dataset cannot be employed for training and testing herein, it is essential to re-cluster to generate anchor boxes that are suitable for the dataset. Since there exist sensitivity issues for initial clustering center selection and outliers for k-means clustering algorithm, an anchor boxes generation method named k-median + + clustering algorithm is presented.

Similar to the clustering algorithm in YOLOv4, Intersection over Union (IoU) is introduced as the distance in the improved k-median + + clustering algorithm, as expressed in Equations (11) and (12).
(11)$$IoU=\frac{area(C)\cap area(G)}{area(C)\cup area(G)}$$
(12)d=1−IoU
where, *C* is the prediction border, *G* donates the actual border, and *d* represents the distance.

To resolve the sensitivity issue of outliers, the median value of sample points is deployed to update clustering centers.

For the sensitivity issue of initial clustering centers, an initial clustering center *c* is randomly chosen. The distance *D* among each sample point *x_i_* and the existing initial clustering centers can be calculated to acquire the probability *P* that the sample point can be regarded as the next initial cluster center, as expressed in Equation (13).
(13)$$P=\frac{D{\left(x_{i}\right)}^{2}}{\sum_{i=1}^{n}D{\left(x_{i}\right)}^{2}}$$

The next initial clustering center is determined by roulette method and k initial clustering centers are obtained. Through continuous iterative update of clustering centers, the appropriate anchor boxes are generated.

### 3.4. Data Augmentation

Mosaic data augmentation is employed in YOLOv4, which splices four images after flipping, scaling and color gamut transformation to enrich samples and ameliorate the generalization and robustness of the model. Since the detection and recognition of traffic lights mainly focus on colors and digitals, the original Mosaic data augmentation is not appropriate. Hence, the flipping and color gamut transformation modules of the original Mosaic data augmentation are deleted. The random difference and other modules are added, which are more suitable for the detection and recognition of colors and digitals. Moreover, Copy-Paste data augmentation [32] and Mixup data augmentation [33] are applied. Copy-Paste data augmentation is adopted to enhance detection the effect for small targets. Mixup data augmentation is employed to further enrich the backgrounds of targets. In this paper, Copy-Paste data augmentation is first applied to data samples, then Mosaic data augmentation is applied on enhanced samples, and finally Mixup data augmentation is employed to the second enhanced samples. To avoid excessive semantic gaps of the sample after three times’ data augmentations, the use probability of three data augmentations is adjusted in combination with actual testing data. Three data augmentation methods are fused to expand sample size and enhance generalization and robustness of the model.

## 4. Results of Evaluation Criteria

To evaluate overall performances of different algorithms, Average Precision (AP), mean Average Precision (mAP), model size, model parameters, floating point of operations (FLOPs) and detection speed are regarded as evaluation criteria.
(14)$$Precision=\frac{TP}{TP+FP}$$
(15)$$Recall=\frac{TP}{TP+FN}$$
(16)$$AP=\int_{0}^{1}Precision\cdot d(Recall)$$
(17)$$mAP=\frac{\sum_{i=1}^{n}AP}{N}$$
where, *Precision* denotes precision rate, Recall represents recall rate, *TP* is the number of correctly identified positive samples, *FP* stands for the number of incorrectly identified positive samples, and *FN* is representative of the number of incorrectly identified negative samples.

## 5. Dataset Test

### 5.1. Test Environment

The test operating system is Ubuntu 18.04, the GPU is NVIDA GeForce RTX 3090 24G, and the CPU is Intel i9-11900K 3.5 GHz. The test environment is CUDA 11.2, cuDNN 8.4.0, Python 3.8. The deep learning framework is Pytorch 1.7.0.

### 5.2. Dataset

The open-source traffic lights dataset named Small Traffic signal light Dataset (S^2^TLD) with 5786 images from Shanghai Jiao Tong University is selected, including 14130 instances of five categories consisting of red, yellow, green, off and waiting. Some typical images are displayed in Figure 7.

Meanwhile, 2872 traffic lights images have also been collected in Chongqing, China, in total containing 7862 instances of 12 categories consisting of red, green, red LED light digitals within five and green LED light digitals within five. The self-made dataset involving a rich environment is possessed of superior quality, with 1247 LED light digitals, which are more suitable for actual autonomous driving conditions. It provides long-term meaning for subsequent relevant researches. Some of the images are revealed in Figure 8.

### 5.3. S^2^TLD Dataset Test

#### 5.3.1. The Effects of K-Median++

The S^2^TLD dataset is clustered, and the clustering effects are presented in Figure 9.

The average IoU of clustering algorithms are listed in Table 3.

YOLOv4 is utilized as the main body for training and testing. Epoch and batch_size are set to 150 and 16, respectively. The mAP of the trained model is obtained and the results are shown in Table 4.

As can be seen from Table 3 and Table 4, compared with the k-means clustering algorithm in YOLOv4, the k-median++ clustering algorithm utilized herein increases by 2.12% the average IoU and by 1.51% the mAP. The above-mentioned prove that the k-median++ clustering algorithm is equipped with better accuracy for generating anchor boxes.

#### 5.3.2. The Effects of CS^2^A Attention Mechanism

The CS^2^A attention mechanism proposed is based on the CA attention mechanism, which also performs better on lightweight networks. Thus, ShuffleNetv2-YOLOv4 is selected as the subject for training and testing. Epoch and batch_size are set to 150 and 16, respectively. The mAP of the trained model is obtained and the results are revealed in Table 5. It can be perceived that compared with the CA attention mechanism and the CBAM attention mechanism, the CS^2^A attention mechanism proposed herein increases the mAP by 4.70% and 6.05%, respectively. Conclusions can be drawn that the proposed CS^2^A attention mechanism is superior to the CA attention mechanism and the CBAM attention mechanism on a lightweight network, which significantly enhances overall accuracy of the model.

#### 5.3.3. Ablation Test

To prove the effectiveness and reliability of each module of the proposed algorithm, YOLOv4 is adopted as the main body for the ablation test. Epoch and batch_size are set to 150 and 16, accordingly. The mAP of the trained model can be achieved and the results are displayed in Table 6. As can be seen from Table 6, compared with the original YOLOv4 algorithm, the mAP of the improved algorithm increases by 8.04% and the model size decreases by 81.97%. The results confirm that each module of the proposed algorithm is effective and reliable to boost overall accuracy, which caters to the need for smaller size and higher precision in actual autonomous driving.

### 5.4. Self-Made Datasets Test

The self-made dataset is selected for training and testing. Epoch and batch_size are set to 200 and 16, respectively. YOLOv4 and the similar lightweight networks YOLOv4-tiny [34] and MobileNetv3-YOLOv4 [35] are selected for comparison. The mAP of the trained models can be obtained and the results are indicated in Table 7. As can be seen from Table 7, compared with YOLOv4, the model size of the proposed algorithm decreases by 81.97% and the mAP of the proposed algorithm increases by 1.79%. Compared with YOLOv4-tiny, the model size augments by 47.72%, though the mAP increases by 22.28%. Compared with MobileNetv3-YOLOv4, the model size decreases by 18.52% and the mAP increases by 37.2%.

The accuracy comparison of each class can be acquired and shown in Figure 10. It can be seen that compared with YOLOv4, YOLOv4-tiny and MobileNetv3-YOLOv4, the AP of traffic light colors recognition increases by 4.5%, 2% and 2.5%, respectively, and the AP of traffic light digitals recognition increases by 1.1%, 26.1% and 44%, respectively, with the adoption of proposed algorithm. It can be seen that the proposed algorithm has high accuracy in recognizing the state of traffic lights and satisfies the requirements of being lightweight.

Then, the confidence threshold is set to 0.5. The predicted confidence results are elaborated in Figure 11.

It can be seen from Figure 11 that YOLOv4 and YOLOv4-tiny are possessed of high predicted confidence for single targets, although there exists a high miss ratio for these two algorithms. MobileNetv3-YOLOv4 performs poorly in both predicted confidence and miss ratio. The proposed algorithm is equipped with both high predicted confidence and lower miss ratio, which bears better fitting ability and would be more suitable for actual detection.

### 5.5. Analyses of the Results of the Dataset Test

Compared with YOLOv4, the recognition accuracy for traffic light colors of the proposed algorithm increases by 8.04% with the S^2^TLD dataset. The recognition accuracy for traffic light colors increases by 4.5%, the recognition accuracy for traffic light digitals increases by 1.1% and the overall recognition accuracy increases by 1.79% with the self-made dataset. Compared with the similar lightweight algorithms YOLOv4-tiny and MobileNetv3-YOLOv4, the recognition accuracy for traffic light colors increases by 2% and 2.5%, the recognition accuracy for traffic light digitals increases by 26.1% and 44%, and the overall recognition accuracy increases by 22.28% and 37.2%, respectively.

Meanwhile, compared with YOLOv4, the model size decreases by 81.97%, which is more appropriate to satisfy the requirements of a small model and high precision in practical applications.

The results indicate that the proposed algorithm is superior to YOLOv4 and the similar lightweight algorithms YOLOv4-tiny and MobileNetv3-YOLOv4 in the detection and recognition of traffic lights’ state information, having a lower miss ratio and better fitting ability. Compared with YOLOv4-tiny, the model size is augmented but the detection effect is significantly improved, which is more advantageous in practical applications.

## 6. Analyses of the Complexity of the Model

To ensure the application of the proposed algorithm in actual autonomous driving scenarios, it is essential to analyze the complexity of the model. YOLOv4 and the lightweight networks YOLOv4-tiny and MobileNetv3-YOLOv4 are selected for comparison.

The results are displayed in Table 8. It can be perceived that compared with YOLOv4, MobileNetv3-YOLOv4 and YOLOv4-tiny, the FLOPs decrease by 93.87%, 65.16% and 46.70%, respectively. Compared with YOLOv4 and MobileNetv3-YOLOv4, the parameters decrease by 83.13% and 14.42%, respectively.

The conclusion can be drawn that the proposed algorithm is equipped with fewer parameters, fewer calculations, and lower model complexity, which satisfies the demands of actual autonomous driving.

## 7. Real-Vehicle Test

### 7.1. Real-Vehicle Test Environment and Results

To verify the effectiveness and reliability of the proposed algorithm in practical application, an appropriate scene is selected for a real-vehicle test. The test operating system is Ubuntu 18.04, the GPU is NVIDA GeForce RTX 3090 24G, and the CPU is Intel i9-11900K 3.5 GHz. The test environment is CUDA 11.2, cuDNN 8.4.0, Python 3.8. The deep learning framework is Pytorch 1.7.0. The test platform is indicated in Figure 12.

The distance of accurate recognition is taken as one of the evaluation criteria to prove the superiority of the proposed algorithm in actual detection and recognition. It can be concluded that as other factors remain unchanged, the recognition effect for small targets is correlated positively with accurate recognition distance. This means that more response and decision-making time is available for the autonomous vehicle.

The models trained in Chapter 5.4 are introduced and the confidence threshold is set to 0.5 for detection and recognition. The location of stable and continuous detection boxes is recorded, as described in Figure 13.

As can be perceived from Figure 13, the distance that the proposed algorithm achieves accurate and stable detection is the farthest, followed by YOLOv4 and MobileNetv3-YOLOv4. YOLOv4-tiny has the worst effect.

The distance of detection and recognition is demonstrated in Table 9. It can be derived that the proposed algorithm has the farthest accurate recognition distance as neglecting the measurement error. Compared with YOLOv4, the detection distance increases by 37%. Compared with YOLOv4-tiny and MobileNetv3-YOLOv4, the detection distance increases by 45.7% and 42.5%, respectively. It can be concluded that the proposed algorithm is more effective in recognizing small targets and more accords with the requirements of actual autonomous driving.

The detection speed can be obtained in Table 10. The conclusion can be drawn that the detection speed of the proposed algorithm increases by 16% compared with YOLOv4.

### 7.2. Analyses of the Results of the Real-Vehicle Test

The real-vehicle test results indicate that the image detection speed increases by 16% and the recognition effect for small targets increases by 37% compared with YOLOv4 in the condition of neglecting other errors. Compared with YOLOv4-tiny and MobileNetv3-YOLOv4, even though the detection speed decreases, the recognition effect for small targets increases by 45.7% and 42.5%, respectively. In an overall view, it can be seen that the proposed algorithm is possessed of better real-time performance and higher accuracy in actual autonomous driving conditions.

## 8. Conclusions

To guarantee fast and accurate recognition of traffic lights by autonomous vehicles in urban traffic, a traffic lights detection and recognition method based on improved YOLOv4 is proposed. Through the introduction of a lightweight network, the optimization of the generation of anchor boxes, the adding of an attention mechanism and data augmentation, both recognition accuracy and a lightweight model can be ensured. 

The main contributions can be summarized as follows:(1)A new method for generating anchor boxes is adopted herein. Compared with the traditional k-means clustering algorithm, the k-median ++ clustering algorithm is less sensitive to outliers and initial values.(2)A novel attention mechanism is propounded herein. A spatial attention mechanism tightly coupled with channel attention further loosely couples with spatial attention, which makes the new attention module pay more attention to scale space information while making a compromise between scale space and feature channel to boost the recognition effect for small targets in a complex background.(3)A multiple data augmentation fusion method is adopted. Three data augmentation methods are fused and the hyper-parameters are adjusted by actual test to expand sample size and improve the generalization and robustness of the model.(4)Compared with YOLOv4, the proposed algorithm has better recognition effect for the state of traffic lights. The model size is significantly reduced, the detection speed is increased, the fitting ability is stronger and the recognition effect for small targets is better, which corresponds to the detection demands of actual autonomous vehicles for traffic lights. Compared with the similar lightweight algorithms YOLOv4-tiny and MobileNetv3-YOLOv4, the proposed algorithm has a slight decrease in how lightweight it is. Nevertheless, the effect on actual target detection can be significantly enhanced, and it is more suitable for target detection in actual autonomous driving.(5)The established traffic light dataset was artificially collected outdoors and contains 2872 images, 12 categories and 7862 examples, including sunny, cloudy, night, evening and other different lighting conditions scenes, as well as red, green, red LED light digitals within five and green LED light digitals within five. Rich target and superior image quality provide the foundation for subsequent research on the perception of traffic lights.

However, the overall mAP of the models is not high owing to limited digital image collection, and future work will focus on the expanded self-made dataset. Meanwhile, in consideration of the poor fitting ability of YOLOv4, a network with better fitting ability can be selected by utilizing XAI tools [36] to enhance or further optimize it to resolve the issue of the poor fitting ability of YOLOv4. Since the detection and recognition of traffic lights in actual autonomous driving is essentially the detection of small targets in complex backgrounds, deeper research on the detection of small targets will be conducted to improve the effectiveness of algorithms in target detection under the conditions of actual autonomous driving. In addition, a novel network can be designed by inputting images and vehicle motion information simultaneously to provide a new method for the target detection of actual autonomous driving [37].

## Figures and Tables

**Figure 1 sensors-22-07787-f001:**
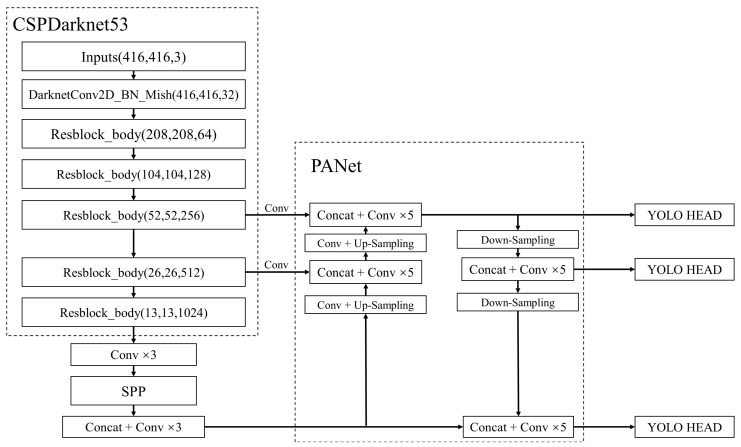
The overall network structure of YOLOv4.

**Figure 2 sensors-22-07787-f002:**
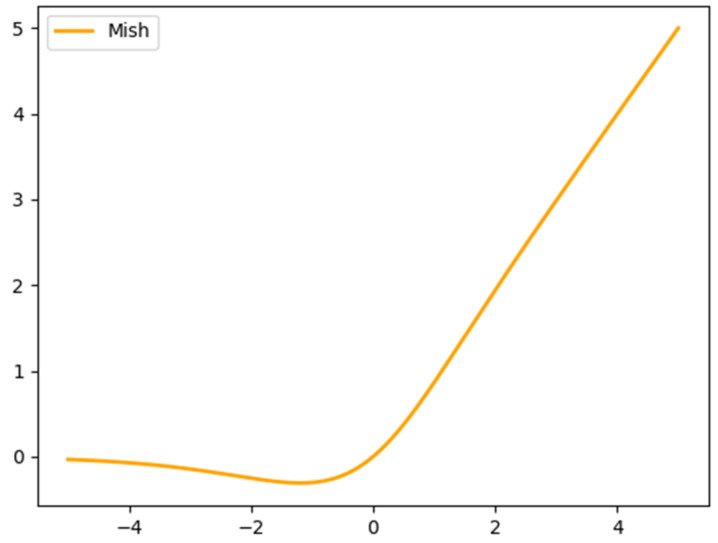
The function graph of Mish.

**Figure 3 sensors-22-07787-f003:**
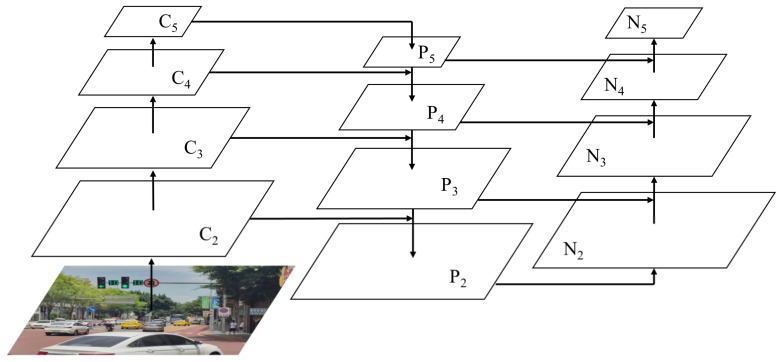
Network structure of PANet.

**Figure 4 sensors-22-07787-f004:**
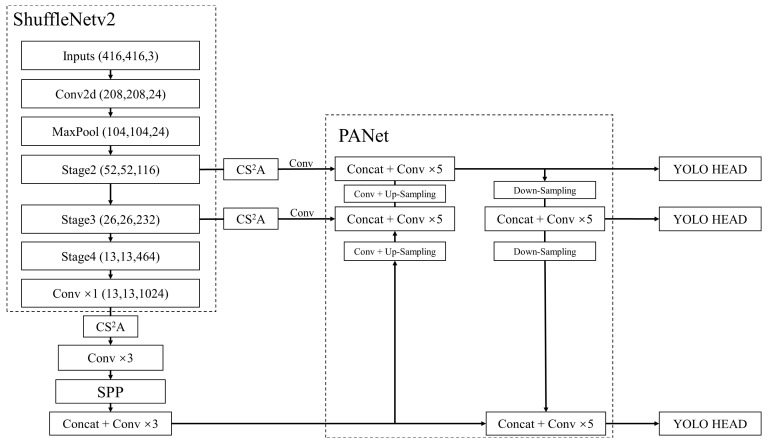
Overall structure of improved YOLOv4.

**Figure 5 sensors-22-07787-f005:**
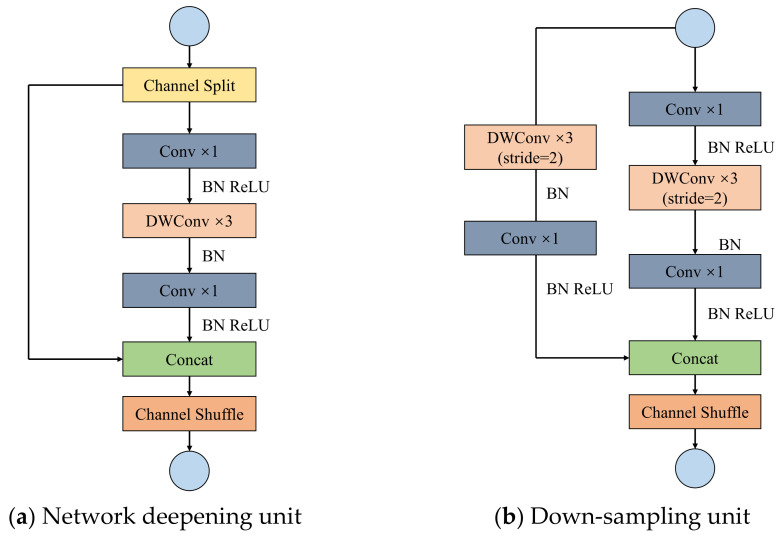
Two types of block unit for ShuffleNetv2.

**Figure 6 sensors-22-07787-f006:**
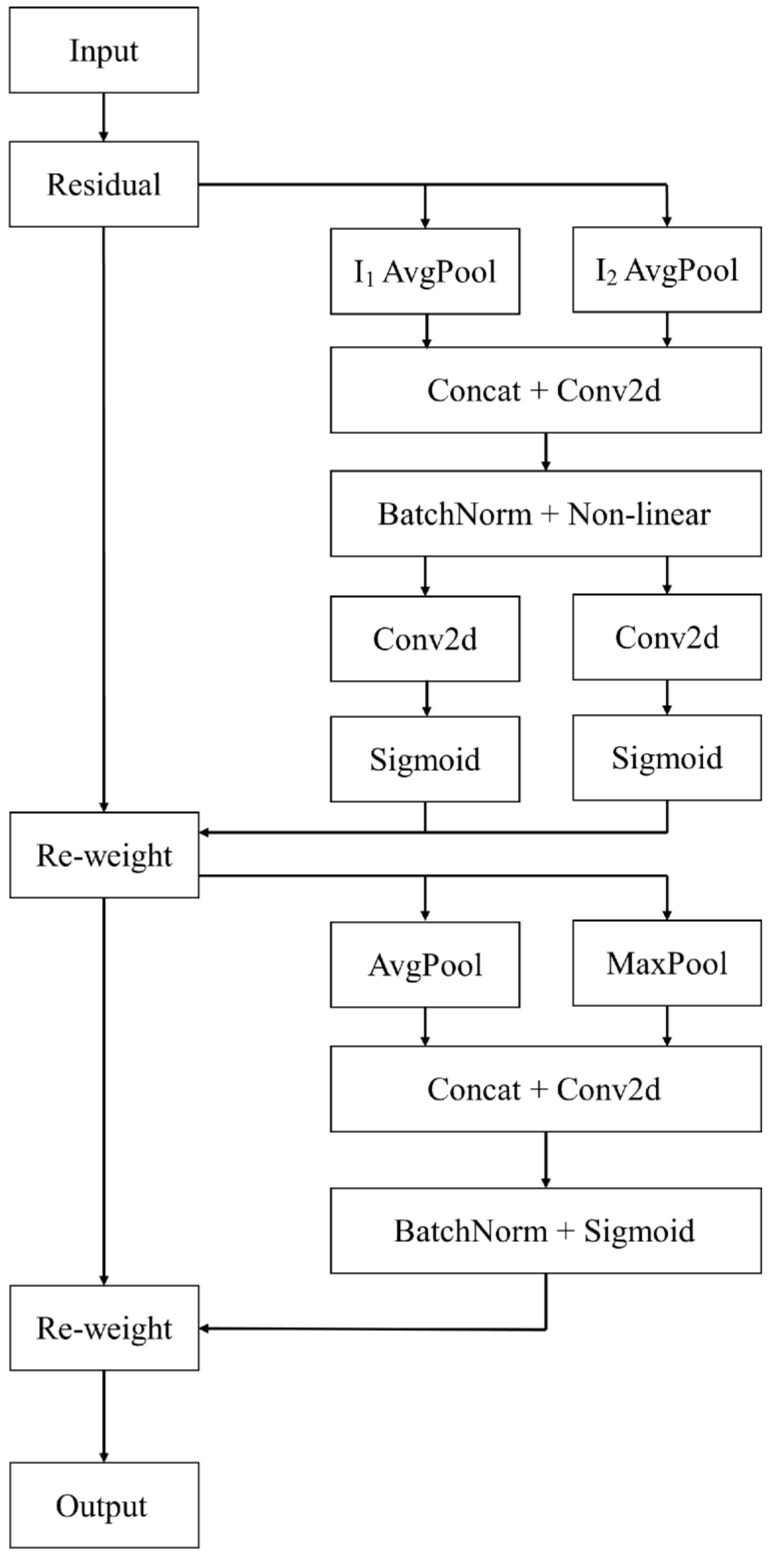
The schematic diagram of CS^2^A.

**Figure 7 sensors-22-07787-f007:**
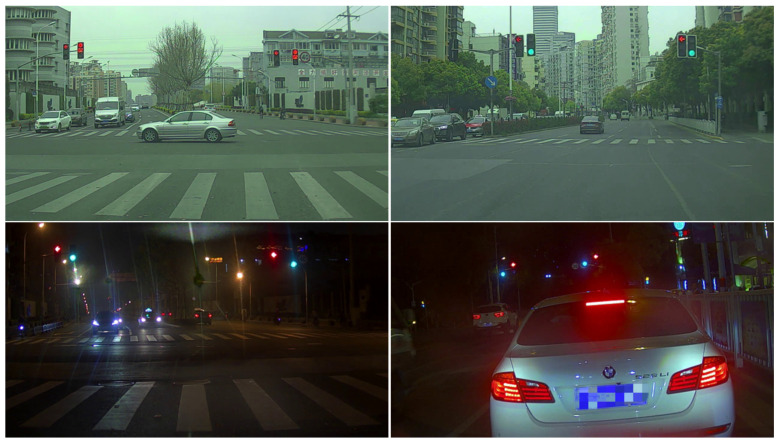
Example images of S^2^TLD dataset.

**Figure 8 sensors-22-07787-f008:**
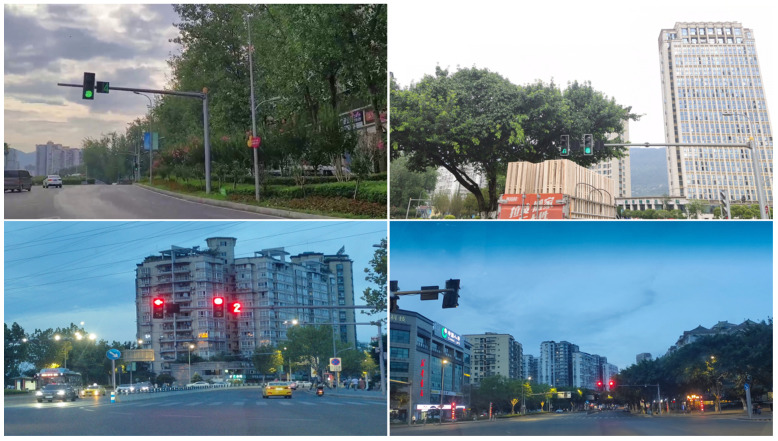
Example images of self-made dataset.

**Figure 9 sensors-22-07787-f009:**
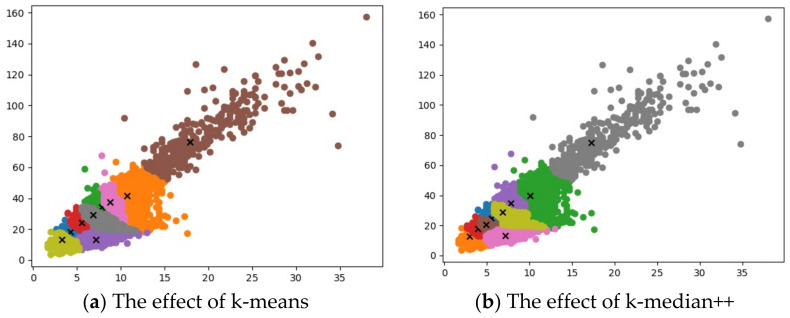
The effects of clustering algorithms.

**Figure 10 sensors-22-07787-f010:**
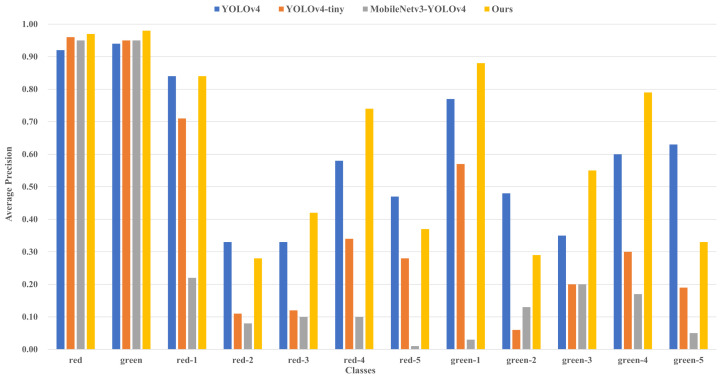
The accuracy comparison of each class in different algorithms.

**Figure 11 sensors-22-07787-f011:**
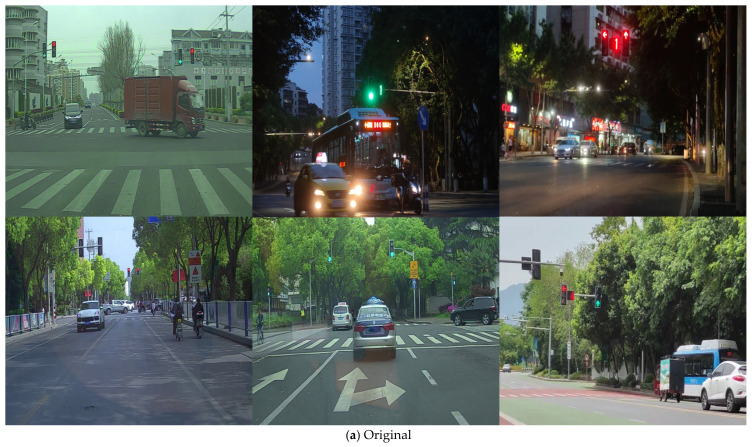
The predicted confidence results of different algorithms.

**Figure 12 sensors-22-07787-f012:**
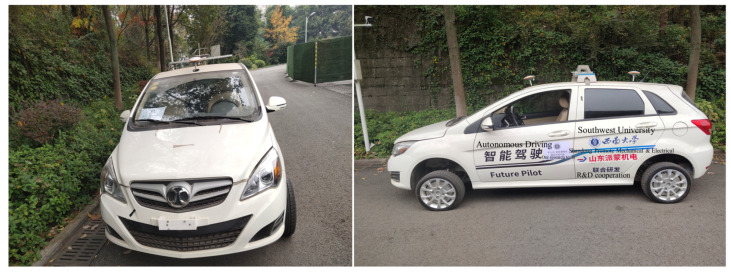
Test platform.

**Figure 13 sensors-22-07787-f013:**
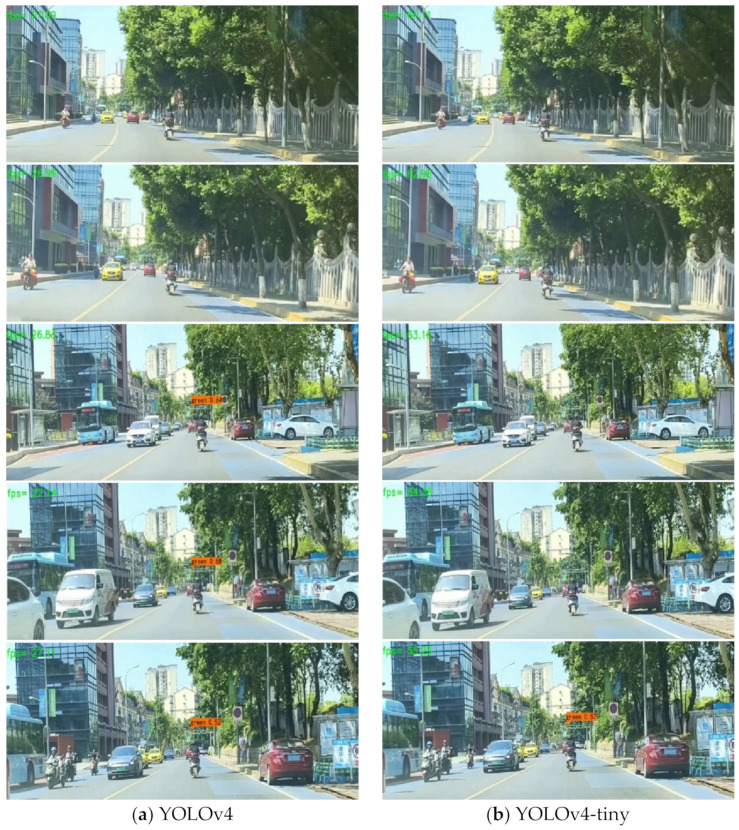
Comparisons of real-vehicle test results for different algorithms.

**Table 1 sensors-22-07787-t001:** Main reviewed works related to traffic light recognition strategy and lightweight/small target.

Traffic light recognition strategy and algorithm improvement	Ref. [11]	Fast RCNN; Genetic optimization
	Ref. [12]	YOLOv2; ShuffleNetv2; Batch normalization layer; Nonlinear activation function layer
	Ref. [15]	YOLOv3; Leapfrog feature fusion; K-means
	Ref. [16]	YOLOv4; Shallow feature enhancement mechanism; Bounding box uncertainty prediction mechanism
	Ref. [17]	LeNet-5; Swish activation function; Dropout regularization
	Ref. [19]	TLRNet; Invalid background removal; Adaptive edge detection
Lightweight/Small target algorithm improvement	Ref. [13]	YOLOv3; DenseNet; Space pyramid pooling
	Ref. [14]	Mixed depth separation convolution; Super-lightweight spatial attention mechanism; GhostNet
	Ref. [18]	GhostNet; YOLO; EAC attention mechanism; Distance-IOU loss
	Ref. [20]	Generative adversarial network; CA attention mechanism

**Table 2 sensors-22-07787-t002:** Overall structure of ShuffleNetv2.

Layer	Size	Stride	Channels
Input	416 × 416	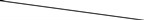	3
Conv2d	208 × 208	2	24
MaxPool	104 × 104	2	24
Stage2	52 × 52	2	116
52 × 52	1
Stage3	26 × 26	2	232
26 × 26	1
Stage4	13 × 13	2	464
13 × 13	1
Conv ×1	13 × 13	1	1024
GlobalPool	1 × 1	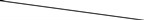	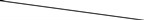

**Table 3 sensors-22-07787-t003:** Clustering results.

Clustering Algorithm	k-Means	k-Median++
Avg_IoU	85.63%	87.75%

**Table 4 sensors-22-07787-t004:** Training results 1.

Clustering Algorithm	k-Means	k-Median++
mAP_0.5_	63.20%	64.71%

**Table 5 sensors-22-07787-t005:** Training results 2.

Attention Mechanism	CA	CBAM	CS^2^A
mAP_0.5_	66.54%	65.19%	71.24%

**Table 6 sensors-22-07787-t006:** Results of ablation test.

YOLOv4	k-Median++	ShuffleNetv2	Mix Data Augmentation	CS^2^A	mAP_0.5_	Model Size
√					63.20%	244 MB
√	√				64.71%	244 MB
√	√	√			61.03%	43 MB
√	√	√	√		66.83%	43 MB
√	√	√	√	√	71.24%	44 MB

**Table 7 sensors-22-07787-t007:** Test results of self-made dataset.

	YOLOv4	YOLOv4-Tiny	MobileNetv3-YOLOv4	Ours
mAP_0.5_	60.33%	39.84%	24.92%	62.12%
Model Size	244 MB	23 MB	54 MB	44 MB

**Table 8 sensors-22-07787-t008:** Results of complexity comparison.

	YOLOv4	YOLOv4-Tiny	MobileNetv3-YOLOv4	Ours
Parameters	64.36 M	6.06 M	12.69 M	10.86 M
FLOPs	60.52 G	6.96 G	10.65 G	3.71 G

**Table 9 sensors-22-07787-t009:** Distance of recognition and detection.

	YOLOv4	YOLOv4-Tiny	MobileNetv3-YOLOv4	Ours
Distance	138 m	119 m	126 m	219 m

**Table 10 sensors-22-07787-t010:** Detection speed.

	YOLOv4	YOLOv4-Tiny	MobileNetv3-YOLOv4	Ours
FPS	26.51	61.14	33.75	31.55

## Data Availability

Not applicable.

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
