# Peer review of "Study on Detection and Recognition of Traffic Lights Based on Improved YOLOv4"

_sensors, 2022, doi:10.3390/s22207787_

Round 1
Reviewer 1 Report
1) Probably the abstract is a little bit too long and could be compressed without sacrificing clarity.
2) The statement of contributions should be rephrased aiming at a more structured exposition (e.g. possibly in a bullet list form). Additionally, the relevant technical challenges tackled by the authors should be better explained.
3) Please complement the related work discussion with a table categorizing the reviewed works along their main distinctive characteristics so as to better highlight/underline the novelty provided.
4) The authors should provide a wider context to the application of DL to sensor systems, including the application to fault detection and identification:
"Nuclear power plant thermocouple sensor-fault detection and classification using deep learning and generalized likelihood ratio test." IEEE Transactions on nuclear science 64.6 (2017): 1526-1534.
"Sensor-fault detection, isolation and accommodation for digital twins via modular data-driven architecture." IEEE Sensors Journal 21.4 (2020): 4827-4838.
5) Please add a paper organization paragraph at the end of Sec. I.
6) The authors should discuss the computational complexity involved in the YoloV4 architecture and the additional computational burden incurred when adding the reported improvement.
7) It is not clear to me whether the considered dataset is (will be) made publicly available by the authors. This would help fostering reproducibility and further advancements on the topic.
8) At the end of conclusion section, the paragraph on future research directions may be enriched more. Indeed, one interesting option is the use of XAI tools to interpret (and possibly improve) the considered network, e.g.:
"Adadi, Amina, and Mohammed Berrada. "Peeking inside the black-box: a survey on explainable artificial intelligence (XAI)." IEEE access 6 (2018): 52138-52160.
Reviewer 2 Report
In this paper, the traffic light detection and recognition method for autonomous vehicles is presented, which is based on improved YOLOv4. The topic addressed in the paper is potentially relevant to the readers of the journal. The overall structure of the paper is good, however, some issues need to be resolved before publication.
In the Introduction section, the authors should provide a clear difference between their work with the reference [18]. In addition, the motivation of their work is not clear i.e., what were the limitations of existing works and how does their work tackle these issues?
Some references are not complete, such as references [20] and [30], i.e., missing journal information.
It is written that "The models trained in Chapter 4.4 are introduced", where is Chapter 4.4?
CA term should be expanded in the introduction section where it was first introduced.
Some grammar errors need to be resolved.
Reviewer 3 Report
Comments:
This manuscript presents a traffic lights recognition method based on improved YOLOv4, to overcome low detection accuracy of small targets and the difficulties to detect and recognize traffic lights in real time and accurately. The research is within the scope of the journal, and I have following general comments:
Comment 1: The justification for the main idea of this paper is insufficient.
Comment 2: The related work does not seem to have much relevance to the proposed scheme.
Comment 3: There is not enough content in the experimental chapter and result part. The justification of this experiment and the description for results are insufficient. The description of the experimental environment is also insufficient.
Comment 4: Refer to MDPI's reference form and rewrite it.
Comment 5: Paragraph breaks are too frequent. The overall paper needs to be arranged.
Comment 6: The paper must be reorganized and re-organized as a whole. Sentences are too long.
Comment 7: No clearly defined methodology for the article. Were these review studies, literature review or desk research? Please complete this.
Round 2
Reviewer 1 Report
The authors have satisfactorily addressed my previous comments and modified their manuscript accordingly.
Reviewer 3 Report
Paper is improved and authors has incorporated all my suggestions in the paper. Paper can be accepted after addressing following comments:
1. Fig. 3: What initial empty blocks without any name represents in the diagram. It needs to be completed.
2. Paper English check is needed.
